# Garcinol as an Epigenetic Modulator: Mechanisms of Anti-Cancer Activity and Therapeutic Potential

**DOI:** 10.3390/ijms262210917

**Published:** 2025-11-11

**Authors:** Geethika Pochana, Tejaswini Sai Karanam, Shacoya Mack, Balasubramanyam Karanam

**Affiliations:** Department of Biology and Cancer Research, Tuskegee University, 1200 W Montgomery Rd, Tuskegee, AL 36088, USA

**Keywords:** garcinol, *Garcinia indica*, *Garcinia cambogia*, cambogin, cancer, histone acetyltransferases, polyisoprenylated benzophenone, apoptosis

## Abstract

The limitations of conventional cancer therapies, including toxicity and resistance, underscore the need for safer and more versatile alternatives that can either complement or substitute existing regimens. Garcinol, a polyisoprenylated benzophenone derived primarily from the rind and leaves of *Garcinia indica* and *Garcinia cambogia*, has drawn significant interest in recent decades. Although traditionally used to relieve inflammatory disorders, its biomedical relevance expanded considerably after reports in the late 20th century demonstrated antimicrobial and subsequently anti-cancer properties. A growing body of cell-based research, supported by a smaller set of animal experiments, now suggests that garcinol acts as a potent epigenetic modulator. Its activities include inhibition of histone acetyltransferases (HATs), a groundbreaking research followed by regulation of oncogenic microRNAs, and modulation of signaling pathways critical to tumor progression. This review integrates current findings on garcinol’s dual role as a HAT inhibitor and regulator of oncogenic networks with updates on in vitro and in vivo studies with a more focused approach on in vivo animal models, highlighting its potential as an emerging therapeutic against malignancies and inflammatory diseases. Nonetheless, translation into clinical settings remains premature, as robust in vivo evidence is sparse and human trials are lacking. Moving forward, systematic investigations are required to clarify safety profiles, establish effective dosing strategies, and evaluate its efficacy across different cancer types.

## 1. Introduction

*Garcinia cambogia* (commonly known as Malabar tamarind) and *Garcinia indica* (Kokum) are tropical fruits native to India and Southeast Asia, valued for centuries in traditional medicine for their anti-inflammatory properties.

Today, extracts from *G. cambogia* are widely marketed as dietary supplements, particularly in weight management. Phytochemical analyses reveal that the rinds of these fruits are abundant in organic acids, amino acids, xanthones, benzophenones, and flavonoids. Among these constituents, garcinol—present at approximately 2–3% of rind content—has emerged as a key bioactive molecule, first structurally characterized by Sahu and colleagues [1]. The standard extraction process involves sequential treatments: initial water extraction to remove hydroxycitric acid, followed by exhaustive methanolic extraction of the residue, solvent partitioning, and final purification [2,3].

Structurally, garcinol shares several functional similarities with curcumin. The C-3 ketonic group and hydroxylated phenolic rings participate in redox reactions, while the α,β-unsaturated ketone moiety facilitates apoptosis induction [Figure 1C and Figure 2]. Antioxidant activity is conferred by double bonds within the isoprenyl side chain, whereas the dihydroxy substituents at carbons 13 and 14, together with the C8 side chain, have been implicated in its cytotoxic effects against cancer cells [4,5].

## 2. Mechanisms of Anti-Cancer Activity

The recognition of garcinol as a cell-permeable histone acetyltransferase (HAT) inhibitor [7] provided a molecular explanation for its anti-cancer activity that had been noted in earlier studies [8,9,10]. The major outcomes of garcinol’s ability to cause epigenetic modulation of cancer are depicted in Figure 3. This seminal finding not only clarified garcinol’s mechanism of action but also positioned it as a key epigenetic modulator capable of altering chromatin dynamics and transcriptional control. By inhibiting HAT enzymes such as p300/CBP and PCAF, garcinol affects the acetylation status of multiple transcription factors and histones, thereby regulating a broad range of oncogenic and tumor-suppressive pathways. Subsequent studies have shown that this modulation extends to critical signaling cascades, including NF-κB, STAT3, PI3K/AKT, MAPK, and Wnt/β-catenin, resulting in the suppression of inflammation, angiogenesis, proliferation, and metastasis. Moreover, its HAT-inhibitory activity has been linked to the regulation of microRNAs, epithelial–mesenchymal transition (EMT), and cancer stem cell renewal, establishing garcinol as a multifaceted compound that integrates epigenetic control with transcriptional and signaling network regulation in cancer.

In the pioneering work on HeLa cells [7], garcinol suppressed the expression of more than 70% of evaluated genes, many of which govern cell division, programmed death, and oncogenic processes. Since then, anti-cancer effects have been documented across multiple malignancies—including ovarian, gastric, colon, oral squamous, glioblastoma, hepatocellular, melanoma, breast, pancreatic, and endometrial cancers—underscoring its broad relevance [11,12,13,14,15,16,17,18,19,20,21,22,23,24]. The present review focuses on garcinol’s mechanisms in cancer prevention and therapy, as evidenced in both in vitro and limited in vivo contexts, while excluding studies related solely to obesity, infectious diseases, and dietary supplementation for livestock production.

Subsequent extensive studies revealed that garcinol regulates diverse intracellular pathways through complex interactions with various molecules involved in the cancer development and its progress. Molecular pathways included but are not limited to cell cycle arrest and apoptosis, HAT inhibition, DNA methylation, histone acetylation, downregulation of oncogenic miRNAs, upregulation of tumor-suppressive miRNAs, inhibition of NF-kB, STAT 3, P13/Akt, COX-2, MAPK pathways, inhibition of epithelial-to-mesenchymal transition (EMT), and cancer stem cell targeting. These pathways are illustrated in Figure 4.

Cell-cycle arrest and the apoptosis induction ability of garcinol have been widely documented across various tumor models. It exerts a dual regulatory influence on cancer cell proliferation and survival by halting progression at specific checkpoints and initiating apoptosis. These mechanisms have been demonstrated in numerous in vitro systems and confirmed in a few in vivo animal studies. Garcinol commonly blocks the cell cycle at the S-phase, leading to the suppression of cell division and enhanced apoptotic signaling. At the molecular level, this response is linked to decreased expression of *cyclin D1* and *cyclin D3*, inhibition of cyclin-dependent kinases (*CDK2*, *CDK4*), and downregulation of the PI3K/AKT pathway, which plays a central role in tumor survival and metastatic progression [25,26,27,28,29].

Additionally, garcinol activates tumor-suppressor proteins such as *p53* and inhibits thioredoxin reductase, resulting in elevated intracellular reactive oxygen species (ROS). The accumulation of ROS subsequently stimulates JNK activation and DNA damage signaling, reinforcing *p53* function and promoting apoptosis rather than mere growth inhibition [30]. Studies in MCF-7 breast cancer cells further support this mechanism, showing activation of mitochondrial apoptosis through the ROS/JNK/ATF-2/Bcl-2 axis [31,32]. Structurally related analogs, including epigarcinol and isogarcinol, have exhibited similar effects by increasing ROS levels and inducing mitochondrial dysfunction in leukemia and prostate cancer models [33,34].

In rhabdomyosarcoma cells, garcinol was also shown to trigger endoplasmic reticulum (ER) stress, elevating the expression of stress-responsive genes such as *DDIT3*, *DDIT4*, *TRIB3*, and *SESN2*, which facilitate apoptosis under prolonged stress [35]. Another regulatory mechanism involves upregulation of *p21^Waf1/Cip1*, a *p53*-dependent CDK inhibitor that prevents G1–S phase progression. This pathway was highlighted in lung cancer studies, where garcinol caused apoptosis in *p53*-competent H460 cells and G1-phase arrest in *p53*-deficient H1299 cells [26,36,37].

Overall, the available evidence demonstrates that garcinol suppresses proliferation and promotes apoptosis through multifaceted signaling interactions, as summarized in Table 1, which lists its key molecular targets and associated outcomes.

Although garcinol is shown to halt cell cycle and promote apoptosis across diverse cancer models, several mechanistic, contextual, and translational gaps persist, including lack of clarity on molecular hierarchy and chromatin context. The hypotheses for the future could be that garcinol induces a bifurcation point where prolonged HAT inhibition stabilizes p21-dependent arrest, and persistent chromatin stress subsequently activates intrinsic apoptotic signaling via mitochondrial depolarization. Whether secondary signaling pathways (PI3K/Akt, MAPK, STAT3) also participate apart from HAT inhibition by garcinol in causing cell cycle arrest and apoptosis needs investigation. Systematic dissection of these unresolved questions will clarify whether garcinol can serve as a prototype epigenetic cell cycle/apoptosis modulator suitable for clinical development.

The NF-κB signaling pathway is known to be constitutively activated in cancer cells, which is a hallmark of numerous malignancies. Suppression of this pathway represents one of the best-established anti-cancer mechanisms of garcinol. The NF-κB network governs the expression of diverse gene families, including pro-inflammatory cytokines (*TNF-α*, *IL-6*, *IL-1β*), anti-apoptotic factors (*Bcl-2*, *Bcl-xL*, *survivin*), and cell cycle regulators such as *cyclin D1*. Garcinol interferes with upstream events that trigger NF-κB activation, leading to reduced nuclear translocation of p65 and consequently a diminished transcription of pro-inflammatory and survival-related genes.

In breast cancer models, garcinol counteracts estrogen-driven proliferation by lowering acetyl-p65 levels and suppressing *cyclin D1*, *Bcl-2*, and *Bcl-xL* expression, while simultaneously activating caspases and other pro-apoptotic mediators—effects largely attributed to the inhibition of NF-κB signaling [38,39,40,41,42]. Comparable outcomes have been reported in hepatocellular carcinoma cells lacking functional p53, where garcinol initiated death-receptor-mediated apoptosis, elevated the Bax/Bcl-2 ratio, and enhanced caspase activity [43]. In glioblastoma (C6) cells, garcinol treatment reduced NF-κB expression and downstream gene products associated with cell survival and proliferation, including Bcl-2, Bcl-xL, survivin, and cyclin D1 [44] and PDGFR [45].

While garcinol has been shown to block IκBα phosphorylation and p65 nuclear translocation, the precise upstream target(s) responsible for this inhibition remains ambiguous. It remains unclear whether NF-κB inhibition is a downstream result of reduced global histone acetylation or a consequence of the specific disruption of p65 acetylation at Lys310. Since NF-kB also plays a role in immunological activation, a complete blockade may be undesirable. Collectively, the emerging hypothesis is that garcinol does not simply inhibit NF-κB signaling, but rather epigenetically rewires the NF-κB transcriptional network through HAT inhibition, redox modulation, and cross-pathway interference. Elucidating these multi-level mechanisms through integrated systems and translational research will be critical for advancing garcinol, its analogs, or combination drugs toward epigenetic-NF-κB-targeted cancer therapeutics.

The microRNA (miRNA) expression is regulated by garcinol through its dual regulatory role, both as an up regulator of tumor-suppressive miRNAs and as a down regulator of oncogenic miRNAs [Figure 5], depending on the targeted molecular pathways in different types of cancers. It shows this dual role both as an epigenetic and transcriptional modulator. Garcinol causes upregulation of the miR-200 family (miR-200b, miR-200c) in breast cancer cells [13] and in drug-resistant non-small cell lung cancer (NSCLC) cells [46] mainly by reversing the epithelial-to-mesenchymal transition (EMT). Also, garcinol upregulates let-c family miRNAs and miR-218 in breast cancer and NSCLC cells by the suppression of EMT and stemness [46]. miR-200c was upregulated in pancreatic cancer stem-like cells by garcinol targeting Notch1/Oct4 resulting in a loss of stemness [14] and miR-181 was upregulated in glioblastoma cells by inhibiting STAT3 resulting in reduced migration/invasion [15].

Oncogenic miRNAs including miR-196a, miR-495, miR-605, miR-638, miR-453, and miR-21 were downregulated in pancreatic adenocarcinoma (PANC-1) cells by garcinol reducing anti-apoptotic signaling (PTEN upregulation and Bcl-2 inhibition) and suppressing proliferation and resistance markers with resulting sensitization to chemotherapeutic agents [47].

Results of studies in medulloblastoma and glioblastoma models demonstrated the suppression of platelet-derived growth factor receptor (PDGFR) and NF-κB pathways, respectively, further supporting garcinol as a pluripotent anti-cancer agent [13,45,46,47]. Overall effects on miRNA networks by garcinol could be summarized as reprograming of miRNA networks toward an anti-cancer phenotype, bridging epigenetic modulation with signaling inhibition and reversal of EMT, loss of stemness, and sensitization to chemotherapeutic agents. Beyond apoptosis, garcinol suppressed metastasis by inhibiting the expression of metalloproteinases (MMPs), enzymes that are involved in tumor resistance, by downregulating miRNAs and attenuating STAT3 and AKT phosphorylation [13,14,25,48,49,50].

The molecular basis for the miRNAs’ regulation by garcinol remains poorly defined. It is unclear whether garcinol directly alters miRNA promoter acetylation through HAT inhibition, influences processing by Dicer/Drosha, key enzymes in the miRNA biogenesis pathway, or affects miRNA stability and export. Integrated analyses linking miRNA shifts to downstream signaling and phenotypic outcomes such as EMT reversal and cancer stem cell (CSC) depletion are absent. A mechanistic understanding of how garcinol reprograms miRNA networks is essential for translating these findings into predictive biomarkers and therapeutic strategies.

Vascular endothelial growth factor (VEGF) is a potent signaling protein that stimulates angiogenesis, such as in cancer, by promoting endothelial cell proliferation, migration, and tube formation. It was shown to support cancer progression by enhancing the supply of oxygen and nutrients to the tumor cells. VEGF expression, proliferation, and colony formation were inhibited in oral squamous carcinoma cells by garcinol without harming normal cells [13,47]. Garcinol has been shown to downregulate VEGF by suppressing NF-kB activity [13], STAT3 [51], and by HAT inhibition. As a p300/PCAF inhibitor, garcinol reduces transcriptional programs that favor angiogenic gene expression. p300 is a HIF-1α co-activator at the VEGF promoter [7]. Garcinol’s anti-angiogenesis effect in cancer by inhibiting VFGF has been demonstrated in vitro [13,19,47,48]. The exact regulatory mechanism for suppression of VEGF and angiogenesis by garcinol remains unclear.

Although garcinol has been reported to downregulate VEGF expression through inhibition of NF-κB, STAT3, and HIF-1α pathways, the precise molecular hierarchy and tissue-specific effects remain inadequately characterized. The temporal kinetics of VEGF suppression, its reversibility, and crosstalk with angiogenic regulators such as VEGFR2, PDGF, and FGF signaling are poorly understood. Moreover, no studies have systematically assessed how garcinol’s epigenetic modulation (e.g., HAT inhibition or miRNA restoration) converges on VEGF gene regulation at chromatin or promoter levels. In vivo data are sparse, and dose-dependent anti-angiogenic effects, pharmacokinetics, and tumor vascular normalization outcomes remain to be clarified. Clinical and therapeutic relevance for a combination of garcinol with other anti-VEGF agents such as bevacizumab, sunitinib, and sorafenib in cancer are yet to be studied for their potential to inhibit VEGF.

Epithelial-to-mesenchymal transition (EMT) is a process in which epithelial cells acquire mesenchymal features such as enhanced motility, invasiveness, and apoptosis resistance and its regulation has been shown as an anti-cancer activity of garcinol. As an epigenetic modulator (p300/CBP/PCAF HAT inhibitor), garcinol targets chromatin and reduces transcription of EMT drivers (e.g., Snail, ZEB, Twist) via reduced acetylation of histones and non-histone factors (NF-κB, STAT3). This epigenetic–EMT link of garcinol-mediated miRNA reprogramming that reverts to MET was summarized in a recent review [14]. In both in vitro and in vivo xenograft models for aggressive breast cancer, garcinol reversed EMT to mesenchymal-to-epithelial transition (MET) with increased E-cadherin and loss of vimentin/ZEB1/2, alongside suppression of Wnt/β-catenin signaling [52]. EMT suppression by garcinol has been shown in esophageal, gastric, colorectal, pancreatic, and liver cancers with anti-metastatic phenotypes and apoptosis, positioning EMT control as a cross-tumor mechanism [19]. The mechanistic pathway of garcinol as an epigenetic EMT-modulator offers translational research avenues using garcinol alone or in combination with other drugs that target EMT while tracking the biomarkers.

The anti-cancer effects of garcinol in melanoma correlated with expression of the adhesion molecule T-cadherin, suggesting that biomarker-driven stratification could identify patients more likely to benefit [53,54]. In breast and osteosarcoma cells, garcinol altered histone acetylation patterns, reducing H3K18 acetylation while increasing H4K16 acetylation and activating DNA damage markers such as γH2A.X [55]. Pancreatic cancer research further suggested that garcinol and its analogs can trigger both apoptosis and pyroptosis, mediated through STAT3 inhibition [56].

Pro-inflammatory cytokine production, such as that of TNF-α, IL-1β, IL-6, and IL-8, was inhibited by garcinol. Mechanistically, this effect is primarily mediated through the inhibition of NF-κB and STAT3 signaling pathways, leading to reduced transcription of cytokine genes. In LPS-stimulated macrophages and cancer cell models, garcinol inhibited IκBα phosphorylation and p65 nuclear translocation, thereby attenuating the NF-κB-dependent expression of TNF-α and IL-6 [8,57]. It also downregulated COX-2 and iNOS, further dampening inflammatory cascades. In addition, HAT inhibition by garcinol contributes to the transcriptional repression of cytokine promoters [7]. In animal models of inflammation and carcinogenesis, treatment with garcinol resulted in decreased systemic levels of IL-6 and TNF-α [58], supporting its role as an epigenetically active anti-inflammatory compound with therapeutic potential. By reducing oxidative stress and inflammatory mediator expression including pro-inflammatory cytokines, garcinol may indirectly modulate immune responses which need to be investigated in in vivo animal models.

Synergistic interactions of garcinol with established chemotherapy drugs have been documented in a few in vitro and in vivo studies. Beneficial effects were observed when combined with conventional chemotherapeutics such as paclitaxel, gemcitabine, and cisplatin by enhancing apoptosis, suppressing NF-κB/STAT3 signaling, and overcoming drug resistance. Garcinol’s combination therapy potential lies in augmenting antitumor efficacy, reducing required drug doses, and minimizing toxicity through complementary targeting of epigenetic and signaling pathways. Garcinol enhanced curcumin-induced apoptosis in pancreatic cancer cells [59], also enhanced TRAIL-mediated apoptosis in renal, lung, and liver cancers [60], and potentiated cisplatin efficacy in ovarian cancer [61]. In pancreatic cancer models, garcinol sensitized cells to gemcitabine through microRNA-dependent mechanisms [46,48,50]. Combined with low-dose Taxol, garcinol significantly enhanced apoptosis and suppressed metastasis in 4T1 breast cancer models [62].

Co-administration of garcinol with synergistic compounds inhibits tumor growth more effectively than monotherapy. When combined with HDAC inhibitors such as SAHA (Suberoylanilide Hydroxamic Acid), also known by its clinical name Vorinostat, garcinol provides complementary effects on maintaining histone acetylation balance, leading to a more effective regulation of gene expression [7,63]. In combination with DNA methyltransferase (DNMT) inhibitors like decitabine, it enables dual targeting of both DNA and histone epigenetic marks, thereby promoting the reactivation of silenced tumor suppressor genes [64,65]. Additionally, when used alongside chemotherapeutic agents such as doxorubicin or cisplatin, garcinol sensitizes resistant tumor cells by restoring apoptotic gene expression and overcoming drug resistance mechanisms [52,66,67].

Combinations have yielded encouraging results showing reduced tumor burden and improved drug sensitivity, suggesting a role for garcinol in reducing chemotherapy dosage and associated side effects. Further research is needed to optimize dosing, understand molecular synergy, assess toxicity profiles, and validate the enhanced therapeutic efficacy in clinical settings.

Cancer stem cells (CSCs) have been described as a small subpopulation of cells within the tumor with self-renewal and pluripotency (via Oct4, Nanog, Sox2), resistance to radiation and chemotherapy, EMT profile, and high possibility for cancer potential. Garcinol has been shown to target CSCs by epigenetic, transcriptional, and signaling modulation, leading to loss of stemness, decreased sphere formation, and restored sensitivity to chemotherapeutic agents. However, inhibition of STAT3 signaling appears to be the key molecular pathway by which garcinol targets CSCs. Garcinol inhibited tumor sphere formation, and reduced NOTCH1 and Wnt/β-catenin signaling, resulting in MET in aggressive breast cancer models [52,68,69,70]. In an in vivo experiment, garcinol suppressed tumor growth in STAT3-activated xenografts, reducing CSC markers [69]. Garcinol inhibits pancreatic cancer stem cell characteristics by modulating microRNA expression and the epithelial–mesenchymal transition [71]. These results suggest that garcinol may not only inhibit proliferating tumor cells but also the self-renewing cells that cause relapse and resistance to chemotherapeutic drugs. In summary, the mechanistic link appears to be that STAT3 normally maintains CSC traits by activating stemness genes and garcinol blocks this transcriptional program through both direct inhibition and reduced acetylation by blocking HAT. Despite promising in vitro and xenograft evidence, critical gaps remain. The pharmacokinetic profile of garcinol, its bioavailability in CSC niches, and the potential for synergistic combinations with chemotherapy or immunotherapy remain poorly defined. Future research should focus on in vivo CSC tracing models, patient-derived organoids, and nanoparticle formulations to improve delivery and target engagement. Integration of multi-omics and single-cell sequencing approaches could further clarify the transcriptional and epigenetic networks through which garcinol eradicates CSCs.

Nanoparticle-based research with garcinol-loaded nanoparticles is showing some promise in cancer therapeutic development. Native garcinol suffers from poor aqueous solubility, rapid metabolism, and limited bioavailability, which restrict its clinical translation despite strong in vitro efficacy. To overcome these limitations, researchers have developed nanoparticle-based formulations—including liposomes, polymeric nanoparticles, solid lipid nanoparticles, and PLGA- or chitosan-based nano-carriers—that encapsulate garcinol for controlled release and tumor-targeted delivery. These nanoformulations have demonstrated enhanced cellular uptake, prolonged plasma half-life, and superior cytotoxicity in breast, colon, and lung cancer models compared with free garcinol. Some systems also enable co-delivery with chemotherapeutics or siRNAs, achieving synergistic inhibition of NF-κB, STAT3, and EMT pathways while minimizing systemic toxicity. Vitamin E TPGS emulsified PLGA nanoparticles loaded with garcinol administered to B16F10 melanoma tumor-bearing mice showed in vivo deposition at the tumor site, confirming the efficacy of the formulation [72]. Garcinol-loaded PLGA nanoparticles were found to improve bioavailability and targeted delivery to the tumor site, addressing challenges related to garcinol’s poor solubility and hydrophobicity [73]. Garcinol (GAR)-loaded cationic nanoliposomes showed antitumor efficacy on B16F10 melanoma cells in vitro and in vivo using a B16F10 tumor xenograft mouse model [41]. A recent study with bovine serum albumin-based nanoparticles encapsulating garcinol showed improved solubility and enhanced cellular uptake, demonstrating better delivery to the cancer cells [74]. Another study with pH-sensitive garcinol-loaded PLGA nanoparticles coated with Eudragit S100 (GAR-PLGA-ES100 NPs) targeted for colonic delivery released it at the higher pH of the colon (pH 7.4), demonstrating controlled release and localization potential [75]. However, more such studies are needed to assess the long-term efficacy, pharmacokinetics, and safety of the garcinol-loaded nanoparticle-based agents.

In silico (computational) studies have been reported on the anti-cancer properties of garcinol. In the in silico studies, ADMET (absorption, distribution, metabolism, excretion, and toxicity) properties are predicted computationally (using tools like SwissADME, pkCSM, or ADMETlab) to screen natural compounds. For garcinol, in silico ADMET analysis helps estimate whether it is drug-like, orally bioavailable, non-toxic, and whether it can cross the blood–brain barrier—crucial for evaluating its potential as a therapeutic agent. A recent review summarizes mechanisms of garcinol in gastrointestinal cancer that could help map computational targets and wet-lab endpoints. This and other published work related to in silico research is summarized with references in Section 4.

## 3. Anti-Cancer Studies in Animal Models

Despite the abundance of in vitro data, relatively few in vivo investigations have been performed. In vivo models remain indispensable for evaluating the therapeutic efficacy, pharmacokinetics, and safety of garcinol. Experiments using different types of laboratory animals, including normal mice and rats, and xenograft or orthotopic tumor models for therapeutic efficacy, provide complementary insights [31,52,59,72,75,76,77,78,79,80,81,82]. In normal rodent models (e.g., AOM/DSS-induced colon carcinogenesis or TPA-induced skin tumorigenesis), garcinol administration has demonstrated suppression of inflammatory mediators (iNOS, COX-2), restoration of antioxidant enzymes (GST, QR, SOD), and reversal of epithelial dysplasia. In xenograft models, such as B16F10 melanoma, MDA-MB-231 breast cancer, or PC-3 prostate carcinoma, garcinol inhibits tumor growth by downregulating NF-κB, STAT3, and PI3K/AKT signaling while inducing apoptosis and reducing the cancer stem cell phenotype.

A key advantage of animal studies is the ability to test different formulations and administration routes—including oral, subcutaneous, intraperitoneal, and nanoparticle-based delivery—to simulate realistic pharmacological exposure. For instance, liposomal or iRGD-targeted garcinol nanoparticles significantly improved tumor accumulation and bioavailability compared with free compounds. Dose ranges between 2.5 and 100 mg/kg body weight have been shown to be effective in reducing tumor burden without systemic toxicity. Such findings validate the pharmacodynamic activity of garcinol and its analogs in a physiological microenvironment, offering critical translational data for clinical development.

However, limitations persist. Garcinol’s poor solubility, rapid metabolism, and variable oral absorption complicate reproducibility across studies. Inter-species differences in metabolism and immune response may also limit direct extrapolation to humans. Moreover, xenograft models often lack an intact immune system, potentially underestimating the compound’s immunomodulatory benefits. Despite these caveats, the convergence of results from multiple in vivo models underscores garcinol’s translational promise as a multi-targeted anti-cancer and chemopreventive agent, warranting further development of optimized analogs and formulations for preclinical and clinical evaluation.

### 3.1. Lymphoma Models

In a primary effusion lymphoma (PEL) model, NOD/SCID mice engrafted with BCBL1-Luc cells received intraperitoneal administration of garcinol at 2.5 and 25 mg/kg every other day for three weeks. Garcinol suppressed tumor growth in a dose-dependent manner [81]. Weaker tumor signal was observed at week 4 post-treatment. At week 8, all garcinol-treated mice showed significantly reduced tumor signals. The specific mechanism involved was not investigated.

In Dalton’s lymphoma, a T-cell lymphoma of spontaneous origin model, Swiss albino mice were injected with Dalton’s ascites lymphoma (DLA) cells. They were orally administered methanol extracts of garcinol from the fruit (GF), bark (GB), and leaves (GL) of *Garcinia morella* at dosages of 100–200 mg/kg body weight for 10 days. Significant increase in median survival time (MST), decreased tumor volume, and restoration of hematological and biochemical parameters were reported [76]. No mechanistic pathways have been reported in these studies.

### 3.2. Liver Cancer—Hepatocellular Carcinoma (HCC)

In xenograft models of liver cancer, athymic nude mice bearing subcutaneous HCC tumors were administered with garcinol at 1–2 mg/kg over four weeks. Tumor volumes were reduced in mice treated with garcinol. Mechanistically, this correlated with inhibition of STAT3 signaling, increased caspase-3 activity, and decreased expression of Bcl-2 [51,82].

### 3.3. Prostate Cancer

Nude mice injected with prostate cancer PC-3 cells (3 × 10^6^) in 200 μL of a PBS solution were used in the study. Mice received an intraperitoneal injection of garcinol (50 mg/kg/d) in 200 μL of corn oil, or orally either 200 μL of corn oil or garcinol in corn oil (50 mg/kg/d) for 5 days per week. Garcinol reduced tumor mass by >80%, with decreased PanIN3 lesions, increased NK cell activity, and increased Caspase-3 and -9, indicating enhanced apoptosis. Cleavage of PARP decreased anti-apoptotic proteins Bcl-2 and Bcl-xL, and increased pro-apoptotic proteins Bax and Bad [79].

### 3.4. Pancreatic Cancer

The KPC transgenic pancreatic cancer mouse model (KPC; K-ras and p53 conditional mutant) was used. Mice were divided into the following groups: KC—Control diet; KGr—0.05% Garcinol diet; KGm—Gemcitabine injected; KGG—Garcinol diet + Gemcitabine injected groups. Dietary garcinol delayed the progression of pancreatic intraepithelial neoplasia and enhanced natural killer cell activity. Combination therapy with gemcitabine further improved outcomes, indicating both direct antitumor and immunomodulatory roles [50]. Results showed improved survival, reduced papilloma formation in the fore-stomach, and increased ratios of NK and NKT cells compared to Non-NK lymphocytes in the KGr group. Tumor sizes were decreased; increases in advanced PanIN3 (high-grade pancreatic intraepithelial neoplasia which is a precursor to invasive ductal carcinoma, IDC) were observed in KGr, KGm, and KGG groups. Garcinol with gemcitabine resulted in decreased tumor volume, increased NK cell activity, and increased survival.

### 3.5. Breast Cancer

Female homozygous ICR SCID mice injected s/c with 5 × 10^6^ MDA-MB-231 cells and used when palpable tumors were observed. Control mice received sesame seed oil only while experimental mice received garcinol by oral gavage (5 mg/d/animal by oral gavage), 6 days per week for 4 weeks. Mice were then sacrificed, and tumors were harvested. Garcinol inhibited NF-κB, vimentin, nuclear β-catenin, and EMT markers, while upregulating miR-200 and let-7 families, reducing tumor vascularization and proliferation [52].

Combined with low-dose Taxol (5 mg/kg, i.p.), garcinol (1 mg/kg, i.g.) was investigated in 4T1 breast cancer models using Balb/c mice injected with metastasis-specific mouse mammary carcinoma 4T1 cells [62]. Garcinol + Taxol enhanced therapeutic efficacy more than garcinol alone. This may be through the induction of Taxol-stimulated G2/M phase arrest and also due to decreased caspase-3/cytosolic Ca^2+^-independent phospholipase A2 (iPLA2) and nuclear factor-kB (NF- B)/Twist-related protein 1 (Twist1) driven downstream events including EMT.

In another study, Balb/c mice nude mice treated with 17β-estradiol and injected with MCF-7 cells (3 × 10^6^ cells per mouse) at mammary sites were used [32]. Following tumor development, the control group received solvent (0.5% DMSO and 0.5% Tween-80 in normal saline) and the treated mice group received cambogin (10 mg/kg in solvent) via i/p injections every other day. Garcinol significantly enhanced apoptosis and suppressed metastasis. At 35 days post-treatment, tumor weight reduced by 72.0% compared with the control group coinciding with increased TUNEL-positive cells, decreased Bcl-2 protein expression, increased Bax protein expression, induction of JNK/SAPK phosphorylation, and AIF nuclear translocation in the tumors of the cambogin-treated mice. Results suggested that cancer development may be, at least in part, through decreased Bcl-2 and activation of JNK/SAPK, both in an AIF-dependent manner.

Garcinol acts as a degradation device to reduce the suppressive activity of regulatory T cells (44) and to enhance the in vivo antitumor activity of a targeted therapeutic anti-p185(her2/neu) (ERBB2) antibody in MMTV-neu transgenics implanted with neu transformed breast tumor cells [83].

### 3.6. Colon Cancer

Five-week-old male Albino ICR mice were used in dextran sodium sulfate (DSS)-induced inflammation and colon carcinogenesis models [84]. Controls were fed a regular diet and the treatment group received a garcinol mixed diet (250 or 500 ppm) for 1 week, followed by 2%, *w*/*v* DSS in drinking water for 14 days. Garcinol prevented DSS-stimulated shortening of the colon length, formation of aberrant crypt foci, reduced inflammation, nitric oxide synthase, COX-2, and proliferating cell nuclear antigen protein expression.

In another colon carcinogenesis model, the groups were as follows: Group 1—control; Groups 2 through 4 received single i/p injection with Azoxymethane (AOM) (10 mg/kg body weight). One week later, Groups 2 through 4 received 2% DSS in drinking water for 7 days. Groups 3 and 4 were fed garcinol (250 or 500 ppm) for 24 weeks. In this model, garcinol decreased tumor size and tumor incidence in the colon, and decreased COX-2, cyclin D1, and VEGF via inhibition of the extracellular signal-regulated protein kinase 1/2, phosphatidylinositol 3 kinase/Akt/p70 ribosomal S6 kinase, and Wnt/β-catenin signaling pathways.

AOM-induced crypt foci were also suppressed in rats fed garcinol diets [85]. Garcinol reduced the proliferating cell nuclear antigen (PCNA) index in ACF, increased liver Glutathione S-Transferases (GSTs-family of phase II detoxification enzymes) and Quinone Reductase (QR) activities, and reduced O^2−^ and NO generation and the expression of iNOS and COX-2 proteins.

In another experiment [86], ten 5-week-old male C57BL/6J mice in each group of six groups were used. High fat diet (HFD) promoted colitis-associated colon cancer as compared to an AOM/DSS group without the intervention of obesity. Garcinol, 0.05% in diet, reduced obesity-promoted colon carcinogenesis. Microbiota of each group was different and clustered.

More recent nanotechnology approaches encapsulating garcinol into PLGA nanoparticles conjugated with targeting peptides improved colon accumulation and survival in colorectal cancer (CRC)-bearing animals. Sprague-Dawley (SD) rats with dimethyl hydrazine (DMH)-induced CRC were fed with garcinol or encapsulated garcinol in biodegradable polymeric nanoparticle (PLGA)-conjugated with iRGD peptide on the particles’ surface (iRGD-GAR-NP’s) [73]. Biodistribution demonstrated the ability of GAR-NP and iRGD-GAR-NP to accumulate in the colons and the iRGD-GAR-NP reduced CRC tumor progression. Survival increased 166% with iRGD-GAR-NP compared to CRC-bearing animals without treatment.

### 3.7. Head and Neck Cancers

Five-week-old male athymic nu/nu mice implanted s/c with human head and neck squamous cell carcinoma (HNSCC) (CAL27) cells (2 × 10^6^ cells/100 μL of saline) were used in the study when tumors reached 0.25 cm in diameter. Garcinol decreased growth and survival of HNSCC by modulating various pro-inflammatory signaling pathways with decreased NF-κB, STAT3, MAPKs, p65, Ki-67, and CD31, and apoptosis induction without showing any toxicity [87].

In another similar HNSCC model study, mice were co-administered garcinol with cisplatin [80]. Tumor growth was decreased with garcinol alone and in combination with cisplatin by downregulating some inflammatory biomarkers. The combination was effective in downregulating the expression of various oncogenic gene products involved in HNSCC growth, survival, invasion, and metastasis along with reduced markers of the proliferation index (Ki-67) and microvessel density (CD31). The combination effectively downregulated oncogenic molecules involved in the proliferation (cyclin D1, COX-2), survival (Bcl-xL, survivin), angiogenesis (VEGF), and invasion (MMP-9, ICAM-1) than with either garcinol or cisplatin and further amplified therapeutic efficacy.

### 3.8. Oral Cancer

Male F344 rats at 7 weeks of age were given 4-Nitroquinoline 1-oxide (4-NQO) at 20 ppm in the drinking water for 8 weeks to induce tongue neoplasms. Dietary garcinol reduced the incidence and multiplicity of 4-NQO-induced tongue neoplasms and/or pre-neoplasms as compared to the control diet, coinciding with reduced BrdU-labeling index, decreased cyclin D1-positive cell ratio, and decreased COX-2 expression, suggesting reduction in cell proliferation activity [88].

Hamster cheek pouch studies similarly showed topical garcinol to reduce DMBA-induced tumors by blocking LTB4 biosynthesis and proliferation [89]. Six to eight-week-old male Syrian golden hamsters were treated with 0.5% 7,12-dimethylbenz[a]anthracene (DMBA) (0.1 mL in mineral oil) topically on the left cheek pouch three times per week for three weeks. Theoretical activity index (J_max_/IC_50_) was highest with garcinol as compared to other compounds used in the study (Zileuton, ABT-761, Licofelone, Curcumin). Therefore, garcinol was predicted as a potent 5-lipoxygenase (5-Lox) pathway inhibitor, which may potentially be used for oral cancer chemoprevention through topical application. Short-term study showed infiltration of inflammatory cells in the submucosa and reduced leukotriene B4 (LTB4) while the long-term study showed decreased tumor volume coinciding with reduced cancer lesions, cell proliferation, and LTB4 and prostaglandin E2 (PGE2) biosynthesis.

### 3.9. Glioblastoma (GBM) and Brain Tumors

NOD/SCID mice inoculated with 1 × 10^6^ U87MG cells (human U87MG glioblastoma cells) in the flank subcutaneously were used in this study [90]. The garcinol-treated group (1 mg/kg body weight, suspended in 0.1% DMSO, i/p) showed reduced size of tumors compared to the control group with no effect on body weights. Garcinol resulted in 100% survival compared to 60% in the control group. Garcinol reduced STAT3, pSTAT3, STAT5A, p-STAT5A, Ki-67, and Bcl-xL expression, while Bax was enhanced and miR-181d expression was enhanced 3.52-fold in the U87MG mice treated with garcinol.

GBM tumor development and growth reduction was suggested due to garcinol abrogating STAT3/5A signaling and upregulating hsa-miR-181d, with concomitant suppression of the Ki-67 proliferation index and enhancement of Bax/Bcl-xL apoptotic ratio [90].

### 3.10. Skin Cancer

CD-1 mice with TPA (12-O-tetradecanoylphorbol-13-acetate)-induced ear inflammation and skin tumors were used [78]. Garcinol was applied topically on the inflamed ear or administered orally. TPA caused the upregulation of pro-inflammatory cytokine IL-6 in the ear skin in a dose-dependent fashion. Oral garcinol reduced UVB-induced ear inflammation and the upregulation of pro-inflammatory cytokine IL-1 beta and IL-6 levels in ears. Garcinol applied to the backs of mice strongly inhibited TPA-induced skin tumor development in mice previously initiated with 7,12-dimethylbenz[a]anthracene (DMBA) with reduced skin tumors/mouse and percent of mice bearing skin tumors.

In a similar study [91], garcinol reduced the TPA-induced expression of inducible nitric oxide synthase (iNOS) and COX-2 and reduced the nuclear translocation of NF-κB and its subsequent DNA binding. This was attributed to blocking the phosphorylation of inhibitor κB α (IκBα) and the p65 subunit. Garcinol also reduced the TPA-induced activation of extracellular signal-regulated kinases (ERK), c-Jun-N-terminal kinases (JNK), p38 mitogen-activated protein kinase (MAPK), and phosphatidylinositol 3-kinase (PI3K)/Akt, which are considered to be molecules upstream of NF-κB [91].

### 3.11. Lung Cancer

Five-week-old NMRI (nu/nu) female mice were injected s/c with A549 cells (2 × 10^7^ in 200 uL of PBS) into the right flank to induce non-small cell lung cancer (NSCLC) [68]. Mice with developed tumors were then injected i/p., with either corn oil (control) or garcinol (15 mg/kg) for 40 days. Tumors in the control group were twice the size of those in the garcinol-treated group.

Garcinol downregulated aldehyde Dehydrogenase 1 Family Member A1 (ALDH1A1) through alterations in the interaction between DNA damage-inducible transcript 3 (DDIT3) and DDIT3-CCAAT-enhancer-binding proteins beta (C/EBPβ), resulting in a decreased binding of C/EBPβ to the endogenous ALDH1A1 promoter. Results suggest that for human patients with NSCLCs showing high ALDH1A1 expression, garcinol could possibly help in reducing this cancer burden [68].

### 3.12. Nanoparticle Delivery Systems

To overcome solubility issues, garcinol has been encapsulated into biodegradable nanoparticles. These formulations demonstrated improved biodistribution, sustained release, and enhanced tumor targeting in vivo, though large-scale efficacy data remain pending [51,72,75].

B16F10 melanoma tumor-bearing mice were injected via tail vein ^99m^Tc-labeled GAR-NPs (0.03 mL, 8–12 MBq/kg). At 2, 4, 8, and 24 h post-injection, blood samples were collected by cardiac puncture of the sacrificed mice. In another study, B16F10 tumor-bearing mice were injected with FITC-labeled NPs, 40 mg/kg body wt. Tumors were collected from mice sacrificed at 2, 4, and 8 h post-injection to study the intra-tumoral distribution of FITC-labeled nanoformulations. Moderate accumulation of ^99m^Tc-labeled GAR-NPs was detected in the tumor region. GAR-NPs also exhibited substantially high cell binding (B16F10 melanoma cells) and internalization indicating improved bioavailability of GAR as a nanoparticulate suspension.

These results combined with oral administration studies [72] focused on the potential antitumor effect of garcinol nanoparticles and support the use of the nanosystem for an effective delivery and anti-cancer efficacy of garcinol.

## 4. Summary and Conclusions

Recent advances have strengthened the view of garcinol as a multifunctional epigenetic modulator with direct and indirect effects on tumor progression. Its capacity to inhibit key histone acetyltransferases such as p300/CBP and PCAF positions it among a new class of natural compounds capable of altering chromatin architecture and transcriptional activity. Beyond histone regulation, garcinol also interferes with the acetylation of non-histone proteins, including transcription factors like STAT3, thereby linking its epigenetic effects to signaling pathways that control cell survival, proliferation, and inflammatory responses. Emerging studies further suggest that garcinol influences microRNA and long noncoding RNA expression, integrating chromatin-based and RNA-mediated control of oncogenic networks. Advances in formulation technologies, including nanoparticle-based delivery, are beginning to address its bioavailability challenges, broadening the scope for preclinical and translational exploration.

However, several critical gaps remain. The selectivity of garcinol for individual HAT isoforms and its precise molecular targets in vivo are not yet fully characterized. Quantitative data on pharmacokinetics, pharmacodynamics, and tissue-specific distribution are limited, hindering a clear understanding of dose–response relationships. Mechanistic connections between garcinol-induced epigenetic changes and downstream modulation of noncoding RNAs still need experimental validation. Comparative studies with other established epigenetic drugs and exploration of potential synergies with immunotherapies or chemotherapeutic agents are also lacking. Finally, while cell-based and animal models provide compelling evidence of efficacy, there is an absence of human data confirming target engagement, safety profiles, and biomarker responses. Addressing these gaps will be essential to translating garcinol’s promising epigenetic activity into clinically meaningful outcomes. Beyond its direct cytotoxic properties, emerging evidence suggests that garcinol may influence the tumor microenvironment through the suppression of angiogenic and inflammatory mediators. Several reports describe decreased levels of pro-inflammatory cytokines such as TNF-α, IL-6, and IL-1β following inhibition of NF-κB signaling, decreased activity of T(reg) cells, and increased NK cell activities. While these findings imply immunomodulatory potential, this area of research is still at an early stage.

Kopytko et al. [14] suggested that histone acetyltransferases (HATs) like p300/CBP support immune evasion by promoting the expression of ligands that dampen antitumor immunity. Since garcinol acts as a natural HAT inhibitor, it may counter this mechanism by enhancing immune recognition. Furthermore, garcinol’s ability to interfere with FOXP3 acetylation could diminish the regulatory T-cell (Treg) suppressive capacity, thereby alleviating immunosuppression within the tumor microenvironment [83]. By downregulating HIF-1α, COX-2, and mPGES-1, key mediators of angiogenesis and immune cell recruitment, garcinol might further shift the tumor microenvironment toward an immune-active state. These actions suggest possible synergy with immune checkpoint blockades, such as anti-PD-1/PD-L1 therapies, which remains an exciting but unexplored avenue. Immune checkpoint proteins, PD-1 (Programmed Cell Death Protein-1), and PD-L1 (Programmed Death Ligand-1), are regulatory molecules that maintain immune homeostasis by either activating or inhibiting immune responses. Tumor cells often exploit these inhibitory checkpoints to suppress antitumor immunity and thus evade immune destruction.

Collectively, the available evidence positions garcinol as a promising candidate for development both as a single-agent therapy and as an adjuvant or sensitizer to existing anti-cancer drugs. Its capacity to influence multiple oncogenic pathways, augment chemotherapeutic responses, and potentially impact chronic inflammatory diseases underscores its value in modern pharmacology. Realizing this potential will require systematic research into its pharmacokinetics, formulation strategies, and safety in clinically relevant models. With sustained interdisciplinary and translational efforts, garcinol could emerge as an important component of next-generation therapeutic regimens, integrating natural product chemistry with precision medicine.

### 4.1. Anti-Metastatic and Anti-Stem Cell Activity

Garcinol’s ability to impede metastasis is a key therapeutic feature. Garcinol downregulates matrix metalloproteinases (MMPs, MMP-2 and MMP-9) as their overexpression in tumors facilitates local invasion, metastatic dissemination, and angiogenesis. Thus, the significance of garcinol’s effect on MMPs lies at the intersection of its anti-metastatic, anti-invasive, and anti-angiogenic properties in cancers. MMP-9 also releases ECM-bound growth factors (VEGF, TGF-β), enhancing tumor vascularization and immune evasion. Therefore, MMP inhibition is a critical anti-metastatic strategy. Broader therapeutic significance is in its dual targeting as other synthetic MMP inhibitors have failed clinically due to toxicity. Garcinol’s multi-target profile (NF-κB, STAT3, HAT inhibition) achieves MMP suppression in a non-toxic, safer way. This aspect needs further investigation. Garcinol-induced inhibition of EMT provides therapeutic potential by reducing metastasis, drug resistance, and cancer stem cell formation.

Emerging evidence also supports garcinol’s efficacy in targeting cancer stem cells, which are often resistant to conventional therapies. By modulating pathways such as NOTCH1 and inducing mesenchymal-to-epithelial transition (MET), garcinol curtails the tumorigenic and self-renewing capacity of these cells. This property may be particularly valuable in aggressive and treatment-resistant cancers like triple-negative breast cancer and pancreatic cancer; however, this aspect needs extensive clinical research.

### 4.2. Insights from In Silico Studies

The very few and recent in silico research publications are providing further molecular insights for garcinol’s anti-cancer activity. In an in silico study, anti-breast cancer potential of Garcinia indica phytocompounds including garcinol was evaluated against nine breast cancer-related protein targets. Molecular docking revealed that garcinol exhibited the strongest binding affinity for BRCA2. ADMET and toxicity analyses confirmed a favorable drug likeness and non-toxic profile suggesting that garcinol may serve as a novel anti-cancer agent [92].

In another study [93], the molecular docking and molecular dynamics of garcinol against the p300/CBP associated factor bromodomain (PCAF Brd) was performed as PCAF is one of the promising target proteins for different types of cancers. This study showed that garcinol forms key interactions and has high binding affinity towards PCAF Brd, suggesting its potential as an anti-cancer therapeutic agent. Further, Rizvi et al. [44] showed through molecular docking analysis that garcinol induces apoptotic cell death by inhibiting NF-kB activity in C6 cells by docking NF-kB through important amino acid residues including Pro^275^, Trp^258^, Glu^225^, and Gly^225^.

More such studies are needed to further delineate and accelerate mechanistic understanding with reduced cost before moving into more in vivo experiments and clinical studies down the line.

### 4.3. Insights from In Vivo Studies

Despite promising in vitro findings, robust in vivo validation is essential. Studies in animal models involving various cancers have shown significant tumor reduction, enhanced apoptosis, and immune modulation. Topical formulations have also demonstrated anti-cancer and anti-inflammatory effects in skin cancer models. Future comprehensive in vivo and clinical studies will help translate its promising in vitro anti-cancer and epigenetic effects into clinically viable treatments, defining optimal formulations, combination drugs, effective drug delivery systems, dosages, and toxicity while targeting different cancer types.

### 4.4. Comparison of Garcinol with Other Known Epigenetic Inhibitors

Garcinol represents a distinctive class of natural epigenetic modulators, functioning primarily as a histone acetyltransferase (HAT) inhibitor that alters chromatin structure and gene transcription. Unlike synthetic HAT inhibitors or well-studied HDAC inhibitors such as vorinostat (SAHA) and trichostatin A, which increase histone acetylation, garcinol exerts the opposite effect, reducing the transcriptional activation of oncogenes. Garcinol complements other natural compounds like curcumin and resveratrol, which exhibit broader epigenetic activity through dual HAT/HDAC modulation. A comparison of garcinol with other known epigenetic modulators is shown in Table 2.

### 4.5. Garcinol Toxicity

Multiple studies in laboratory animals have found garcinol to be well-tolerated with no significant toxic effects including no histological or pathological changes to the liver, kidneys, lungs, heart, or esophagus at the doses used in those studies. No cellular toxicity has been reported in many in vitro studies. In an acute safety study using Wistar rats, 40% Garcinol formulation did not produce any adverse effects at a high single dose of 2000 mg/kg and can therefore be classified as GHS Category 5 (unclassified) according to the Globally Harmonized System (GHS) for chemical classification. Furthermore, no treatment-related changes were observed at the highest tested dose of 100 mg/kg/day in the 28-day and 90-day repeated-dose oral toxicity studies, as well as in the reproductive and developmental toxicity assessments [105]. These findings indicate that 40% Garcinol exhibits a low toxicity profile in rodents and showed no observable adverse effects under the experimental conditions tested. Recently, in a clinical trial with human patients with non-alcoholic steatohepatitis, the clinical efficacy and safety of the garcinol, curcuminoids, and piperine (GCP) combination was investigated. Over a 90-day period, no adverse toxicity effects were observed with no significant changes in hematological and clinical laboratory parameters [106].

While the current evidence provides a favorable indication of garcinol’s safety, definitive conclusions require more rigorous and comprehensive investigations. Future studies should encompass long-term human clinical trials and detailed mechanistic toxicity assessments to elucidate its pharmacokinetics, biodistribution, and potential interactions with co-administered therapeutic agents, particularly chemotherapeutic drugs. It is also imperative to evaluate safety parameters across different age groups, sexes, and physiological conditions, including pregnancy and lactation. Furthermore, systematic assessments of toxicity to different body systems and dose–response relationships should be conducted for various formulations and routes of administration. Such multifaceted studies are essential to establish a robust and clinically relevant safety profile for garcinol under diverse therapeutic contexts.

### 4.6. Advances in Nanotechnology

Nanoparticle-based garcinol formulations are likely to improve bioavailability and solubility as they are hydrophobic with poor aqueous solubility and low bioavailability. Although nanoformulation approaches have substantially improved the solubility and stability of garcinol, several critical gaps remain before clinical translation can be realized. Most current studies are confined to in vitro characterization, with limited in vivo validation of pharmacokinetics, biodistribution, and therapeutic efficacy. Systematic evaluations of toxicity, long-term safety, and optimal dosing strategies are also lacking.

Future research should prioritize in vivo tumor models to confirm therapeutic benefits and elucidate mechanisms of biodistribution and clearance. The scale-up of nanoparticle synthesis under reproducible and regulatory-compliant conditions will be a challenge that must be addressed for translational feasibility. Functionalization of nano-carriers with targeting ligands or antibodies could enhance tumor or immune cell specificity, while co-delivery systems combining garcinol with chemotherapeutic or immunomodulatory agents may further improve efficacy. Ultimately, well-designed preclinical and early-phase clinical trials are needed to establish the safety, pharmacodynamics, and therapeutic potential of garcinol nanoformulations in human cancer management.

### 4.7. Analogs of Garcinol

Despite garcinol’s potent pleiotropic anti-cancer activity, its clinical translation remains limited by poor aqueous solubility, metabolic instability, and limited target selectivity. These drawbacks underscore the need to design novel analogs and derivatives that retain the benzophenone backbone while improving pharmacokinetics and pharmacodynamics. Structural modifications, such as altering the prenyl side chains, hydroxyl positions, and phenolic substitutions, can enhance HAT selectivity, bioavailability, and tumor targeting, offering a rational path toward next-generation epigenetic therapeutics. Synthetic chemistry efforts have generated few semi-synthetic and fully synthetic garcinol analogs aimed at optimizing potency and reducing off-target effects. Table 3 shown below compares the structure, targets, advantages, and translational limitations of natural analogs (Isogarcinol, Epigarcinol, and Camboginol) with the semi and fully synthetic analogs.

To accelerate translation, future work with garcinol analogs should aim for the following: pursue structure–activity relationship studies linking specific phenolic substitutions to HAT vs. HDAC inhibition profiles; in silico computational docking to predict binding affinities for p300, CBP, and STAT3, guiding analog selection; develop pharmacokinetically optimized analogs (single or combination) validated through in vivo bioavailability and safety studies; and evaluate biomarkers of efficacy, including histone acetylation levels, miRNA signatures, EMT reversal, and CSC elimination, to enable patient stratification and precision-guided therapy.

### 4.8. Beyond Oncology: A Broader Therapeutic Horizon

Garcinol’s anti-inflammatory and antioxidant properties extend its potential applications to chronic inflammatory conditions, neurodegenerative diseases, and metabolic disorders such as obesity and diabetes. These prospects align with the rising demand for plant-based therapeutics that are both effective and sustainable.

Furthermore, the exploration of garcinol highlights the broader importance of underutilized medicinal plants in drug discovery and development. As interest in phytochemicals grows, garcinol stands as a compelling candidate for integrative and precision medicine approaches.

### 4.9. Limitations and Future Directions

Although initial in vivo investigations highlight garcinol’s potential as an anti-cancer agent across several tumor models, the available data are not yet sufficient to define standardized dosing protocols or to clarify its complete mechanistic profile. Despite promising preclinical outcomes, clinical translation remains in its infancy. The current evidence base is dominated by in vitro studies, with only limited animal experiments and an absence of human clinical trials. Differences in extraction techniques, formulation strategies, and dosing parameters further complicate the comparison of study results. Consequently, while the mechanistic rationale for garcinol is strong, its therapeutic application in clinical settings remains largely theoretical. Bridging these knowledge gaps through systematic in vivo and clinical research is essential to move garcinol toward practical medical use.

Administration Routes and Dosage Optimization

The comparative efficacy and safety of oral versus intravenous delivery require rigorous evaluation to determine the optimal administration routes and therapeutic dosages with minimal toxicity.

2.Pharmacokinetics and Bioavailability

Systematic studies are needed to characterize absorption, metabolism, and tissue distribution. Nanoparticle formulations and conjugation strategies may help overcome solubility and stability challenges. Effects on immunological parameters need to be expanded to delineate changes in circulatory immune cell subsets, pro- and anti-inflammatory cytokines, and changes in immune checkpoint proteins.

3.Toxicity and Safety Profiles

Comprehensive preclinical safety assessments across different animal models will be necessary to establish tolerability and identify any potential off-target effects.

4.Standardization of Extraction and Purification

Variations in source material and preparation methods make reproducibility difficult. Development of standardized protocols will be important for consistency in experimental and clinical settings.

5.Expanded In Vivo Models

Further exploration in genetically engineered and patient-derived xenograft models would provide more clinically relevant insights into efficacy across different tumor types and stages.

6.Combination Therapies

Given its synergistic effects with agents such as cisplatin, Taxol, gemcitabine, and TRAIL, systematic evaluation of garcinol as an adjuvant therapy could identify contexts where it enhances existing regimens while reducing required drug dosages.

7.Clinical Translation

Early-phase clinical trials should be prioritized once safety and pharmacokinetics are established, with an initial focus on cancers for which strong preclinical evidence already exists (e.g., pancreatic, breast, and colon cancers).

8.Effects on immunological parameters

Experiments devoted to studying garcinol’s immunomodulatory effects are almost nonexistent or very limited. Investigations are needed to delineate changes in circulatory immune cell subsets and their functions, pro- and anti-inflammatory cytokines, and the status of immune checkpoint proteins, PD-1 and PD-L1.

## Figures and Tables

**Figure 1 ijms-26-10917-f001:**
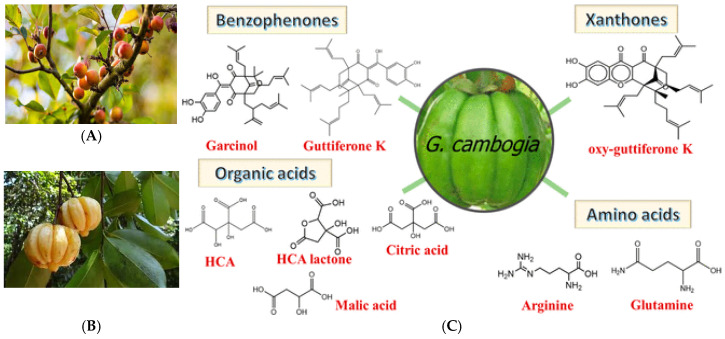
(**A**) *Garcinia Indica*, (**B**) *Garcinia Cambogia*, (**C**) chemical constituents of Garcinia fruit [6].

**Figure 2 ijms-26-10917-f002:**
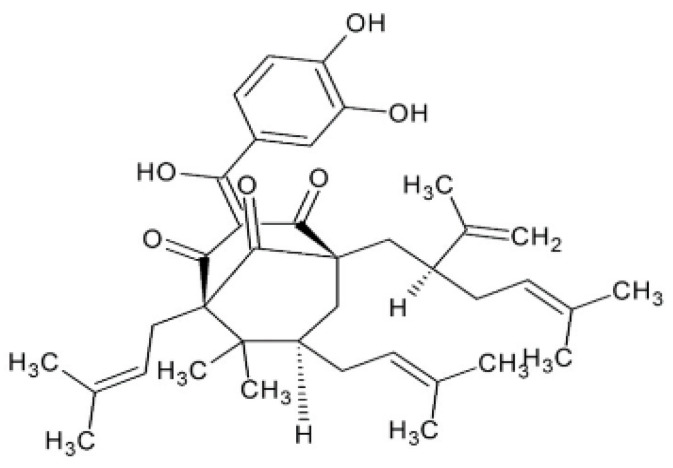
Chemical structure of garcinol.

**Figure 3 ijms-26-10917-f003:**
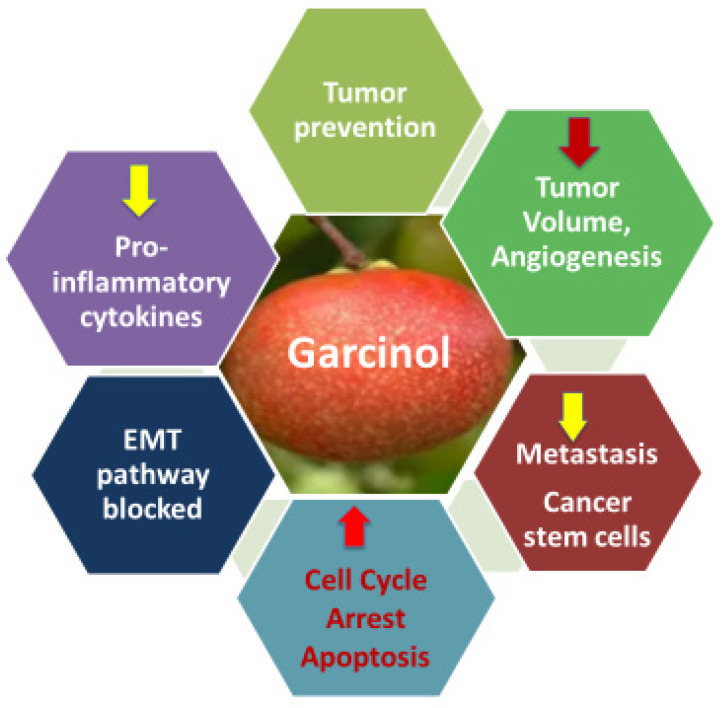
Outcomes of epigenetic modulation and multifunctional anti-cancer effects of garcinol.

**Figure 4 ijms-26-10917-f004:**
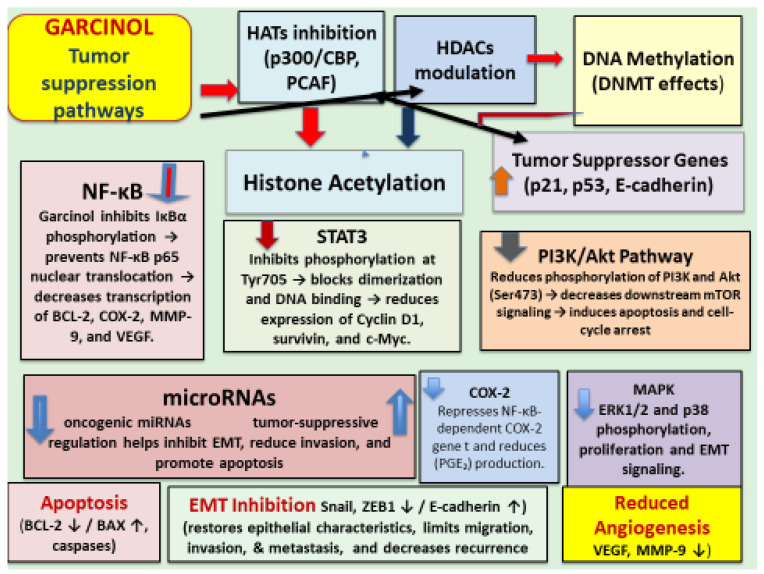
Schematic illustration of the multi-targeted tumor-suppressive mechanisms of garcinol across epigenetic, transcriptional, and signaling axes.

**Figure 5 ijms-26-10917-f005:**
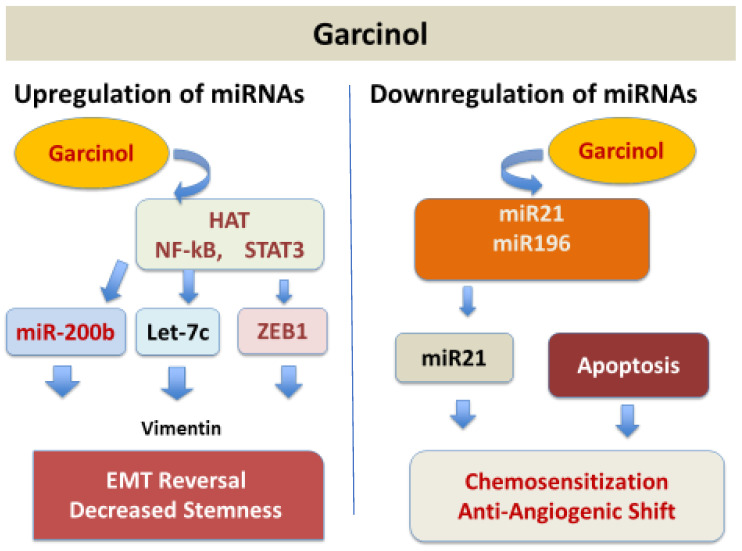
Garcinol upregulates tumor suppressor miRNAs and downregulates oncogenic miRNAs.

**Table 1 ijms-26-10917-t001:** Summary of molecular targets and outcomes involved in cell cycle arrest and apoptosis.

Mechanistic Target	Effect of Garcinol	Outcome
Cyclin D1/CDK4 (G1 progression)	↓ Expression	G1 arrest
p21, p27 (CDK inhibitors)	↑ Expression	Cell cycle inhibition
Bcl-2 family	↓ Anti-apoptotic, ↑ Pro-apoptotic	Mitochondrial apoptosis
Caspases (3, 8, 9)	Activation	Apoptotic execution
NFκB,STAT3,PI3K/AKT	Inhibition	Suppressed survival signaling
HAT activity (p300/CBP) Histone acetylation	Inhibition	Epigenetic repression of oncogenes

**Table 2 ijms-26-10917-t002:** Comparison of epigenetic targets, mechanism of action and cancer relevance of garcinol with other known epigenetic modulators.

Drug References	Epigenetic Target	Mechanism of Action	Effect on Gene Expression	Notes/Cancer Relevance
Garcinol [7,94]	HATs (p300/CBP, PCAF)	Inhibits histone acetylation → chromatin condensation	Represses oncogene transcription (e.g., c-Myc, Cyclin D1)	Anti-inflammatory, pro-apoptotic, antioxidant; natural compound
Curcumin [64,95]	HATs, HDACs, DNMT1 (weak)	Dual inhibitor (HAT/HDAC modulation)	Restores tumor suppressor genes (e.g., p21, p53)	Multi-targeted nutraceutical; lower potency
Vorinostat (SAHA) [96,97]	HDACs (Class I and II)	Inhibits deacetylation → chromatin relaxation	Promotes apoptosis and differentiation	FDA-approved for cutaneous T-cell lymphoma
Romidepsin [98,99]	HDAC1/2	Similar to SAHA	Induces apoptosis via reactivation of silenced genes	FDA-approved; more potent but with side effects
Azacitidine/Decitabine [100,101]	DNMT1	Incorporates into DNA → inhibits methylation	Reactivates silenced tumor suppressor genes	FDA-approved for myelodysplastic syndromes
TSA (Trichostatin A) [102,103]	HDACs	Inhibits deacetylation	Activates transcription of silenced genes	Research use; not clinically approved
Resveratrol [63,104]	SIRT1 (HDAC III)	Activates SIRT1 → promotes deacetylation	Modulates metabolism and longevity pathways	Indirect epigenetic modulation

**Table 3 ijms-26-10917-t003:** Comparison of natural and synthetic garcinol analogs.

Analog Type/References	Representative Compound(s)	Primary Molecular Targets/Pathways	Advantages	Limitations
Natural Analogs [15,107]	Isogarcinol (Cambogin, Camboginol)	NF-κB, COX-2, iNOS, p300/CBP HATs	Naturally occurring isomer with anti-inflammatory and anti-cancer activity; improved metabolic stability over garcinol	Limited potency and solubility; low HAT selectivity
[33]	Epigarcinol	Mitochondrial apoptotic pathway, caspase-3/9	Induces apoptosis and ROS-mediated cytotoxicity in leukemia cells	Poor solubility; limited target validation
[15]	Cambogin/Camboginol	NF-κB, MAPK, PI3K/AKT	Broad anti-inflammatory and antitumor activity in vitro	Non-selective activity; unclear HAT inhibition profile
Semi-Synthetic Derivatives [56]	Hydroxyl/Allyl-modified garcinol derivatives	p300/CBP HATs, STAT3, Cyclin D1, Bcl-2	Enhanced HAT inhibition and apoptosis induction; tunable side chain substitutions	Synthetic complexity; limited in vivo PK data
[18]	Garcinol–Curcumin/Garcinol–Quercetin hybrids	NF-κB, STAT3, HATs, HDACs	Synergistic epigenetic modulation; dual antioxidant and anti-inflammatory effects	Need optimization of linker stability; possible metabolic interactions
Fully Synthetic Analogs [14,56]	Simplified benzophenone scaffolds	p300, PCAF, STAT3, COX-2	Improved solubility, predictable SAR; allows selective design	May lose full biological pleiotropy; limited clinical validation
Nano-enabled Analog Conjugates [18]	PEG/TPGS/iRGD-linked garcinol analogs	Tumor-targeted delivery; sustained release	Enhanced bioavailability and tumor accumulation; reduced systemic toxicity	Stability and scalability challenges; preclinical only

## Data Availability

No new data were created or analyzed in this study. Data sharing is not applicable to this article.

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
