# Peer review of "Garcinol as an Epigenetic Modulator: Mechanisms of Anti-Cancer Activity and Therapeutic Potential"

_ijms, 2025, doi:10.3390/ijms262210917_

Round 1

Reviewer 1 Report

Comments and Suggestions for Authors

The Authors presented a concise and coherent overview of garcinol as a promising bioactive compound with potential therapeutic applications in oncology and inflammation. The authors successfully establish a logical rationale for studying garcinol by emphasizing the limitations of conventional cancer therapies, namely toxicity and resistance and situating garcinol as a safer, multitarget alternative.

A particular strength of the fragment is the integration of mechanistic insights with pharmacological potential. The description of garcinol as both a histone acetyltransferase (HAT) inhibitor and a regulator of oncogenic microRNAs reflects a sophisticated understanding of its epigenetic impact, however, the information provided on this topic is limited to a few sentences concerning HATs. I suggest a more in-depth description of these relationships, even more so since they are included in the title of manuscript.

Authors can also include in silico research. There are few papers regarding these kinds of studies, particularly garcinol, related with cancer that could complete the information in the present manuscript.

Reviewer 2 Report

Comments and Suggestions for Authors

Authors have provided a comprehensive overview of garcinol’s anticancer mechanisms in this review. However, in my opinion the topic of garcinol as an epigenetic anti-cancer agent has been covered in prior literatures. After reading the review, I feel like that authors have compiled majorly known information rather than introducing new hypotheses or perspectives. I think authors should focus more on new headings in the review beyond existing literature on garcinol’s anticancer mechanisms.

You need to focus completely on epigenetic modulation with oncogenic network regulation in this review.

Though you have discussed and cover a wide array of molecular mechanisms, but the in depth analysis for each mechanism is somewhat limited. In many cases, the manuscript lists findings from various studies without deeply analyzing or connecting them.

There is minimal discussion on garcinol’s ability and its role as a HAT inhibitor or an epigenetic modulator. Further, many sections are repeated which are already detailed in the mechanism section.

You need to divide the mechanism section into sub sections like “anti metastatic and anti EMT effects, cancer stem cell targeting, epigenetic effects, microRNA regulation etc….you need to stick to theme of the review. In each sub section then you provide detailed mechanistic insights, rather than presenting one common mechanism.

Very basic figure (Figure 1) showing the source plants (Garcinia indica/cambogia) and the chemical structure of garcinol. This is good for introducing garcinol, but not enough for mechanistic review. In my opinion, the review lacks any schematic or pathway figures that illustrate the complex mechanisms described. You should add at least two schematic figures to represent garcinol’s mechanisms of action showing its effects on various molecular targets. Another figure could focus on epigenetic modulation.

Phrases like “a groundbreaking research of the corresponding author” should be avoided. Self-referential tone needs to be revised.

Part of discussion and final perspective can be merged into a single concluding section.

Some mechanisms are repeated. Please check and remove unnecessary repetition.

There is a lack of clinical and translational discussion on garcinol. Is there any information on garcinol’s toxicity to normal cells or organs in animal models?

You mentioned about immune modulation, but no detail.

Similarly, authors must emphasize on epigenetic modulator part. How garcinol can be compare to other known epigenetic drugs? Please expand this point.

In future directions you can talk about probable emerging areas like nano-formulations or drug delivery, its bioavailability, combinatorial immunotherapy etc……..

I believe if the authors revise the manuscript after implementing on the raised concerns, the review will be significantly improve and of interest to readers.

Round 2

Reviewer 1 Report

Comments and Suggestions for Authors

The authors took into account all the reviewer's comments. 

Reviewer 2 Report

Comments and Suggestions for Authors

Authors need to revise the manuscript again and resubmit the rebuttal properly. This is not the way to submit your responses to the raised concerns. Please address all concerns and respond to each individual comment properly. DO NOT group 12 comments into 3 comments and respond to 3 comments. Currently, authors have responded superficially to the comments I raised, taking only few aspects, merging them together and responding together. Please resubmit the rebuttal again before I re-review the manuscript. 

Round 3

Reviewer 2 Report

Comments and Suggestions for Authors

Some minor corrections are still required:

2. There is minimal discussion on garcinol’s ability and its role as a HAT inhibitor or an epigenetic modulator. 

Where is it you have added the discussion on garcinol’s ability and its role as a HAT inhibitor or an epigenetic modulator???? I cannot see.

4. Why figure 3 legend is mentioned in the figure itself? Please revise the figure. Also, legend is not self-explanatory. It's too short.

5. Part of discussion and final perspective can be merged into a single concluding section. 

You have to understand that there shouldn't be a discussion section in the review article. So, remove the heading discussion. Maintain the continuity of the review and replace the discussion heading with some other heading, end it with conclusion.

Author Response

Please see the attached

Thank you very much for the reviewer’s valuable comments.

We have carefully addressed all the comments provided. As suggested, we have replaced the subheading “DISCUSSION” with “SUMMARY AND CONCLUSIONS”, which we believe is a suitable alternative.

Additionally, we have expanded the discussion regarding garcinol’s properties and its role as a HAT inhibitor and epigenetic modulator. Two new paragraphs have been added to this section (now titled Summary and Conclusions) to provide a more comprehensive explanation of these aspects.
